# Exploration of Antimicrobial Ingredients in *Psoralea corylifolia* L. Seed and Related Mechanism against Methicillin-Resistant *Staphylococcus aureus*

**DOI:** 10.3390/molecules27206952

**Published:** 2022-10-17

**Authors:** Liqiong Sun, Zhijuan Tang, Minxin Wang, Jun Shi, Yajuan Lin, Tiefeng Sun, Zhilu Zou, Zebin Weng

**Affiliations:** 1College of Horticulture, Nanjing Agricultural University, Nanjing 210095, China; 2College of Traditional Chinese Medicine & School of Integrated Chinese and Western Medicine/College of Pharmacy/School Hospital, Nanjing University of Chinese Medicine, No. 138 Xianlin Avenue, Nanjing 210023, China; 3Shandong Academy of Chinese Medicine, Jinan 250014, China; 4School of Basic Medicine, Hubei University of Chinese Medicine, Wuhan 430065, China

**Keywords:** methicillin-resistant *Staphylococcus aureus*, *Psoralea corylifolia* Linn. seed, prenylated flavonoids, UPLC-MS/MS, bacterial membrane

## Abstract

With the abuse of antibiotics, bacterial antibiotic resistance is becoming a major public healthcare issue. Natural plants, especially traditional Chinese herbal medicines, which have antibacterial activity, are important sources for discovering potential bacteriostatic agents. This study aimed to develop a fast and reliable method for screening out antimicrobial compounds targeting the MRSA membrane from *Psoralea corylifolia* Linn. seed. A UPLC-MS/MS method was applied to identify the prenylated flavonoids in major fractions from the extracts of *Psoralea corylifolia* Linn. seed. The broth microdilution method was used to determine the minimum inhibitory concentrations (MICs) of different fractions and compounds. The morphological and ultrastructural changes of MRSA were determined by scanning electron microscopy (SEM). The membrane-targeting mechanism of the active ingredients was explored by membrane integrity assays, membrane fluidity assays, membrane potential assays, ATP, and ROS determination. We identified eight prenylated flavonoids in *Psoralea corylifolia* Linn. seed. The antibacterial activity and mechanism studies showed that this type of compound has a unique destructive effect on MRSA cell membranes and does not result in drug resistance. The results revealed that prenylated flavonoids in *Psoralea corylifolia* Linn. seeds are promising candidates for the development of novel antibiotic agents to combat MRSA-associated infections.

## 1. Introduction

With the abuse of antibiotics, bacterial antibiotic resistance is becoming a major public healthcare issue. Thus, it is urgent to develop antibiotics with novel or multiple targets to overcome the rising challenge of microbial resistance. Methicillin-resistant *Staphylococcus aureus* (MRSA) has become a common pathogen in clinical practice, which is not merely resistant to almost all β-lactam antibiotics, but also resistant to glycopeptides [1]. Recently, more and more attention has been paid to the unexplored MRSA membrane for screening novel potential targets. Antibacterial agents targeting the MRSA membrane have attracted great attention, because those agents could interrupt the membrane integrity, resulting in membrane depolarization, potassium ion efflux, and rapid cell death [2,3]. Currently, bio-guided separation is a commonly used strategy to screen for bioactive compounds such as enzyme inhibitors or agonists from natural plants, while it is still time-consuming and high-cost [4]. Therefore, various high-throughput methods were developed to screen out bioactive compounds quickly and efficiently from natural plants [5,6,7]. However, these methods must be based on known target proteins. It is notoriously difficult to establish high-throughput methods for screening out antimicrobial compounds targeting bacterial membranes, due to the insufficient understanding of potential targets on the bacterial membrane.

Recent studies reported that the isopentenyl flavonoids (IPFs) such as glabrol, licochalcone A, licochalcone C, and isobavachalcone possess potent antibacterial activities against MRSA, which may be because the isopentenyl group could promote compounds attaching to the MRSA membrane [8,9]. Meanwhile, it was also concluded that the IPFs could interact with the major components of cell membranes such as phosphatidylglycerol (PG), phosphatidylethanolamine (PE), and cardiolipin (CL), causing the loss of cell membrane integrity and rapid cell death. We herein hypothesized that the prenylated flavonoids might specifically bind to phospholipids in the bacterial membrane.

Liquid chromatography-mass spectrometry (LC-MS) is one of the most-effective approaches for rapidly identifying ingredients in natural plants. Therefore, many high-throughput screening methods are usually based on LC-MS, such as screening out anti-lipase compounds or acetylcholine binding protein ligands from the extracts of traditional Chinese medicine (TCM) by using ligand fishing coupled with LC-MS [10]. *Psoralea corylifolia* Linn. seed (also known as Buguzhi) is widely used for the treatment of skin infection diseases in China. Recent phytochemical research reported that *P. corylifolia* seed contains abundant IPFs such as corylifol A, bavachin, isobavachin, and bavachalcone [11,12]. In addition, ethanol extract of *P. corylifolia* seed showed promising antibacterial activity against MRSA and *Helicobacter pylori* [13]. However, whether IPFs from *P. corylifolia* seed exert antibacterial activity and their mechanism of action remain unclear. Based on those concepts, we thus aimed to develop a rapid and reliable approach based on the LC-MS methods to screen out the prenylated compounds from *P. corylifolia* seed, which could combat MRSA by targeting the cell membrane.

In the present study, we firstly illustrated the ionic fragments and neutral losses of different types of prenylated flavonoids including 6-prenylnaringenin, licoisoflavone B, glabrone, licochalcone A, etc., in positive ion mode by using UPLC-Q-TOF-MS. According to the related characteristic ions and neutral losses, the major fractions containing prenylated compounds from the extracts of *P. corylifolia* seed were characterized, and the bioassays were then carried out. Compared with the data of standard compounds, major prenylated compounds in the fraction were thus identified. We also confirmed whether those compounds could induce cell death by targeting the bacterial membranes, and the antibacterial actions of the major compounds in *P. corylifolia* seed were demonstrated.

## 2. Materials and Methods

### 2.1. Reagents and Chemicals

The standard compounds of 6-prenylnaringenin, licoisoflavone B, glabrone, licochalcone B, licochalcone D, isobavachin, corylifol A, bavachin, isobavachin, and bavachalcone were purchased from Yuanye Co., Ltd. (Shanghai, China). Dichloro-dihydro-fluorescein diacetate (DCFH-DA) and ATP Assay Kit were purchased from Beyotime (Shanghai, China). 3,3′-dipropylthiadicarbocyanine iodide [DiSC3(5)], propidium iodide (PI), and naphthylamine were purchased from Aladdin (Shanghai, China). Other reagents were all commercially available. The tested strains included *S. aureus* (ATCC 25923), *S. epidermidis* (CMCC 26069), *E. coli* (CICC 20658), and *Shigella dysenteriae* (CICC 23829). MRSA and *Pseudomonas aeruginosa* were two clinically isolated bacteria donated by Nanjing Medical University.

### 2.2. Sample Preparation

The *P. corylifolia* seeds (PCSs) were obtained from Bozhou (Anhui, China). The PCSs were pulverized to a homogeneous size by a grinder. The material powder was extracted three times with 70% ethanol under reflux for 2 h each time to obtain ethanol extracts (EEs). Then, the EEs were suspended in deionized water and partitioned with ethyl acetate (EtOAc) to obtain the EtOAc fraction. The EtOAc fraction was then separated through a silica column eluted with a petroleum ether/EtOAc gradient (6/3, 6/4, 6/5, and 0/1) to obtain four different polarity fractions, named Fractions A–D (Fr. A–D). Fractions A–D were successively used for further research.

### 2.3. Chromatographic Separation and Mass Spectrometry Detection

Chromatography was performed on a Waters^TM^ UPLC system (Waters Corp., Milford, MA, USA). The chromatographic separations were performed on a Waters^TM^ ACQUITYUPLC BEH C_18_ column (2.1 mm × 100 mm, 1.7 μm). The flow rate and column temperature were set at 0.4 mL·min^−1^ and 25 °C, respectively. A mobile phase system consisting of 0.1% formic acid in H_2_O (A)-Acetonitrile (B) was applied with the following gradient program: 0–1 min, 90–70% A; 1–14 min, 70–40% A; 14–18 min, 40–10% A; 18–20 min, 10–90% A. The injection volume was 2 μL.

Mass spectrometry was accomplished on a Waters Synapt™ Q-TOF/MS system (Waters Corp., Milford, MA, USA) equipped with an electrospray ionization (ESI) source in both positive and negative ion mode. The parameters of the ionization source were set as follows: source temperature of 120 °C, capillary voltage of 3 kV, desolvation temperature of 350 °C, sampling cone voltage of 30 V, extraction cone voltage of 2 V. The desolvation and cone gas were both nitrogen and set, respectively, at a flow rate of 800 and 50 L/h. The scan range was between m/z 100 to 1000 Da. The scan time, interval scan time, and collision energy were set, respectively, at 0.5 s, 0.02 s, and 6 eV throughout the whole experiment. Leucine-enkephalin was used as the lock mass, which generates a [M + H]^+^ ion (m/z 556.2771) and [M − H]^−^ ion (m/z 555.2615) in positive and negative modes, respectively. The obtained UPLC-Q-TOF/MS data of all samples were analyzed by the Masslynx software (version 4.2, Waters Corp., Milford, MA, USA) for peak detection and alignment.

### 2.4. Antibacterial Tests

Minimum inhibitory concentrations (MICs) of different fractions and compounds were determined by the broth micro-dilution method according to the CLSI 2021 guideline [14]. Briefly, the compounds or antibiotics were firstly dissolved in dimethylsulphoxide (DMSO) to obtain an initial concentration of 1 mg/mL. Then, the samples were two-fold diluted in Mueller–Hinton broth (MHB) and then mixed with 100 μL of bacterial suspensions containing approximately 1.0 × 10^6^ CFU/mL. The MICs were determined after incubation at 37 °C and 180 r/min for 18 h.

### 2.5. Scanning Electron Microscopy Analysis

The morphological and ultrastructural changes of MRSA were determined by using SEM. Briefly, MRSA cells were treated with the bioactive fraction at final concentrations of 1 MIC and then incubated at 37 °C for 2 h. The culture medium was centrifuged at 5000× *g* for 5 min. The cells were collected and washed with PBS (0.01 M, pH 7.4) three times. The cells were mixed with 2.5% glutaraldehyde for 2 h and washed with PBS (0.01 M, pH 7.4) three times. The cells were dehydrated by using different concentrations of ethanol (30%, 50%, 70%, 90%, and 100%), and those cells were then freeze-dried and coated with gold. The ultrastructure was observed by using a Hitachi SU-8100 field emission microscope.

### 2.6. Antibacterial Activity of the Mixtures of Compounds with Lipids

The phospholipids including PG, PE, and CL were dissolved in 20% DMSO to obtain an initial concentration of 128 μg/mL. The effects of these phospholipids on the antibacterial activity of different compounds were evaluated by using the chequerboard microdilution assay.

Briefly, various phospholipids were two-fold diluted in MHB and then mixed with 100 μL of bacterial suspensions containing approximately 1.0 × 10^6^ CFU/mL. The MICs were determined after incubation at 37 °C and 180 r/min for 18 h.

### 2.7. Membrane Integrity Assays

The effects of different compounds on membrane integrity were evaluated by using a PI fluorescence probe, according to a previous study with some modifications [8]. Briefly, the compounds were firstly diluted in MHB to obtain final concentrations of 1 MIC, 2 MIC, and 4 MIC followed by the addition of bacterial suspensions (OD_600_ = 0.6) and then incubated at 37 °C for 2 h. The cells were collected and washed with sterile PBS (0.01 M, pH 7.4) three times and were then mixed with 10 μL of PI (100 μg/mL) followed by incubation at 37 °C for 30 min without light. After centrifugation at 5000× *g* for 5 min, the cells were washed with sterile PBS (0.01 M, pH 7.4) three times, and the fluorescence was then determined using PerkinElmer 2300 with an excitation wavelength at 535 nm and an emission wavelength at 615 nm.

### 2.8. Membrane Fluidity Assays

The effects of different compounds on membrane fluidity were evaluated by using a laurdan fluorescence probe, according to a previous study with some modifications [15]. Briefly, the cells were mixed with 10 μL of laurdan (150 μg/mL) followed by incubation at 37 °C for 30 min without light. The compounds were then added to obtain final concentrations of 1 MIC followed by incubation at 37 °C for 2 h. After centrifugation at 5000× *g* for 5 min, the cells were washed with sterile PBS (0.01 M, pH 7.4) three times, and the fluorescence value was then determined using PerkinElmer 2300 with an excitation wavelength at 350 nm and an emission wavelength at 440 nm, as well as 490 nm. The laurdan GP value was calculated as the following formula:$$\mathrm{GP}=\frac{\left(I440-I490\right)}{\left(I440+I490\right)}$$
I_440_ represents the fluorescence value where the excitation wavelength is 350 nm and the emission wavelength is 440 nm; I_490_ represents the fluorescence value where the excitation wavelength is 350 nm and the emission wavelength is 490 nm.

### 2.9. Membrane Potential Assays

The effects of different compounds on membrane potential were evaluated by using a DiSC3(5) fluorescent probe, according to a previous study with some modifications [9]. Briefly, the compounds were firstly diluted in MHB to obtain final concentrations of 1 MIC, 2 MIC, and 4 MIC followed by the addition of bacterial suspensions (OD_600_ = 0.6) and then incubated at 37 °C for 2 h. The cells were collected and washed with sterile PBS (0.01 M, pH 7.4) three times and were then mixed with 10 μL of DiSC3(5) (100 μg/mL) followed by incubation at 37 °C for 30 min without light. After centrifugation at 5000× *g* for 5 min, the cells were washed with sterile PBS (0.01 M, pH 7.4) three times, and the fluorescence was then determined using PerkinElmer 2300 with an excitation wavelength at 535 nm and an emission wavelength at 615 nm.

### 2.10. ATP Determination

The effects of different compounds on the extracellular ATP were evaluated by using the ATP Assay Kit, according to a previous study with some modifications [8]. Briefly, the compounds were firstly diluted in MH broth to obtain final concentrations of 1 MIC, 2 MIC, and 4 MIC followed by the addition of bacterial suspensions (OD_600_ = 0.6) and then incubated at 37 °C. After incubation for 2 h, the cells were centrifuged at 5000× *g* for 5 min, and the supernatant was then collected for extracellular ATP determination. Ten microliters of the ATP Assay Kit solution was quickly added to 190 μL of supernatant, and the luminescence value was then determined using PerkinElmer 2300.

### 2.11. ROS Measurement

The effects of different compounds on the intracellular reactive oxygen species (ROSs) were evaluated by using a DCFH-DA fluorescent probe. Briefly, 10 μL of DCFH-DA was firstly incubated with bacterial suspensions (OD_600_ = 0.6) at 37 °C for 30 min without light. After centrifugation at 5000× *g* for 5 min, the cells were washed with sterile PBS (0.01 M, pH 7.4) and were then mixed with different compounds (final concentrations of 1 MIC, 2 MIC, and 4 MIC) followed by incubation at 37 °C. After incubation for 2 h, cells were collected and washed with sterile PBS (0.01 M, pH 7.4) three times, and the fluorescence value was then determined with an excitation wavelength at 488 nm and an emission wavelength at 525 nm.

### 2.12. Resistance Development Studies

MRSA was inoculated in fresh MHB containing 0.5 MIC of different compounds at 37 °C for 18 h, and Oxacillin was used as a positive control. Then, the MIC of these bacteria was determined by using the above method. The cultures were serially passaged for 20 days.

### 2.13. Isothermal Titration Calorimetry Assays

To evaluate the interaction between 1-palmitoyl-2-oleoyl-sn-glycero-3-phospho-(1′-rac-glycerol) sodium salt (POPG) and CLA, the calorimetric experiment was performed by MicroCal ITC at 25 °C. POPG dissolved in HEPES (20 mmol L^−1^, pH 7.0) was sequentially injected into the calorimetric cells filled with CLA, which was dissolved in the same buffer, and the injection was repeated 20 times with an equilibrium interval of 200 s. The processed data were used, the relative software with the instrument calculating the equilibrium dissociation constant (KD), stoichiometry (n), and the changes of enthalpy (ΔH).

### 2.14. Statistical Data Analysis

All parameters detected in this study are presented as the mean ± standard deviation (mean ± SD). The data were plotted using the GraphPad Prism 9.0 software (San Diego, CA, USA). The data were analyzed by one-way ANOVA using the SPSS 23.0 software (Chicago, IL, USA). Statistically significant was defined as a probability (*p*-value) <0.05.

## 3. Results and Discussion

### 3.1. Structural Characterizations of Reference Flavonoids

Since MRSA was discovered, epidemic strains have continued to emerge. MRSA still poses a formidable clinical threat with persistently high morbidity and mortality. Therefore, novel economic medicament agents and various ways are desperately required to tackle the growing drawback of antibiotic resistance [1]. Natural plants are vital resources for discovering drug-lead compounds. However, their applications in the antagonism of clinical pathogens and bacterial resistance are still neglected [16]. Since there are thousands of compounds in natural plants, the key step is to discover the active antimicrobial ingredients by a rational approach. Previous studies have concluded that the ethanol extract from PCSs has significant antibacterial activity [13]. To quickly screen out components with the significant antibacterial activity of PCSs, here, we developed a fast and reliable method for screening out antimicrobial compounds targeting MRSA membranes from PCSs.

Based on previous studies, we knew that most of the flavonoids in PCSs are IPFs [12]. Compared with common flavonoids, isopentenyl flavonoids have special UV absorption and mass spectrometry pyrolysis [17,18]. Therefore, we can quickly pick them out by identifying their characteristic cleavage peaks. We firstly analyzed the pyrolysis law of three different types of IPFs. It was demonstrated that the characteristic neutral losses of 54 Da (C_4_H_6_) and 56 Da (C_4_H_8_) were concerned with the pyran ring or prenyl chain on the substituent moiety. Meanwhile, the characteristic neutral losses of 42 Da (C_3_H_6_), 54 Da (C_4_H_6_), and 70 Da (C_4_H_6_O) were found, owing to the presence of the pyran ring. Based on these characteristic cracking laws and combined with the common characteristic cracking laws of flavanone, chalcone, and isoflavones, we could quickly screen out the IPFs in the EtOAc fraction of the PCSs.

Previous studies reported that the flavonoids in PCSs mainly include isoflavones, chalcones, and flavanones [11,12]. In the present study, different types of reference flavonoids including 6-prenylnaringenin (6-PN), licoisoflavone B (LFB), glabrone (GE), licochalcone B (LCB), and licochalcone D (LCD) were selected to summarize the structural characteristics and MS cracking law of isoflavone, chalcone, and flavanone. As shown in Figure 1 and Appendix A and Table 1, LCB and LCD showed similar dissociation routes in positive ion mode, where the retro-Diels–Alder (RDA) reaction generated ^3^A^+^ fragment ions or ^4^B^+^ as base peaks. Meanwhile, LCD displayed the characteristic neutral losses of 56 Da (C_4_H_8_) and 42 Da (C_3_H_6_) (Figure 1A), due to the presence of a prenyl chain in the A ring. In the ESI ion source, the C ring of flavanone is proven to be opened to form its related chalcone, resulting in the same dissociation routes of flavanone and chalcone [19,20,21]. In this study, 6-PN also generated ^3^A^+^ (Figure 1B) due to the RDA reaction. Furthermore, 6-PN also displayed the characteristic neutral losses of 56 Da (C_4_H_8_), owing to the presence of a prenyl chain in the A ring. As reported in previous studies [19,22,23], the RDA reaction is a common fragmentation pathway for isoflavones, flavanones, chalcones, and isoflavones. Therefore, ^1,3^A^+^ ions were also observed in LFB and GE. Meanwhile, LFB and GE also displayed the characteristic neutral losses of 42 Da (C_3_H_6_), 54 Da (C_4_H_6_), and 70 Da (C_4_H_6_O), owing to the presence of the pyran ring. Taken together, according to the fragments of skeleton moiety degradation containing ^3^A^+^ or ^4^B^+^ ions, it could be predicted to be chalcone or flavanone. Furthermore, the fragments of skeleton moiety degradation containing ^1,3^A^+^ ions could be used to predict the existence of isoflavone. Importantly, the characteristic neutral losses of 54 Da (C_4_H_6_) and 56 Da (C_4_H_8_) corresponded to the pyran ring or prenyl chain on the substituent moiety, respectively.

### 3.2. Identification and Isolation of Compounds in the Fractions of PCSs with Antibacterial Activity

As shown in Table 2, Fr. A possessed potent antibacterial activity against Gram-positive bacteria, and Fr. A was thus used for further study. According to the structural characterizations of reference compounds, the target compounds were firstly chosen (shown in Figure 2A) from Fr. A based on the neutral losses of 54 Da, 56 Da, or 42 Da, which contributed to the prenyl chain or pyran ring (Table 3). The UV data are also useful for determining the skeleton of flavonoids [20]. In terms of UV absorption, compounds **5** and **8** were considered chalcones, which showed strong absorption at 360 nm to 380 nm. Compounds **1**, **3**, and **6** were considered flavanones, which corresponded to the strong absorption at 270 nm to 290 nm. Furthermore, compounds **2**, **4**, and **7** were considered isoflavones, according to the shoulder peak at 280 nm to 320 nm. The accurate molecular weight of those compounds was calculated by mass measurement, and the molecular formula was then obtained by searching the chemical databases such as SciFinder Scholar and PubMed Scholar. Thus, compound **8** was considered as 4′-O-methylbroussochalcone B (MFB), according to its molecular weight and UV absorption. According to the molecular weight and UV absorption of compound **5**, it was considered isobavachalcone (IBC) or bavachalcone. According to the molecular weight of compound **7**, it might be corylifol A (CLA). Meanwhile, it displayed the characteristic neutral losses of 56 Da and 68 Da, as well as the characteristic ions of 137 Da, indicating the presence of two prenyl chains. Therefore, compound **7** was considered CLA. Compound **4** displayed the characteristic neutral losses of 42 Da, 54 Da, and 70 Da, indicating the presence of a pyran ring. Combined with the neutral losses and molecular weight, it was considered corylin. Among compounds **2**, **4**, and **7**, they all showed the characteristic ions of 137 Da, indicating that compound **2** produced the same ion of ^1,3^A^+^. Moreover, it also displayed the characteristic neutral losses of 56 Da, indicating the presence of a prenyl chain. Compound **2** was herein considered as neobavaisoflavone (NBF). Both compounds **1** and **3** showed similar dissociation routes with 6-prenylnaringenin, where compounds **1** and **3** generated the ions of 149 Da, which might be produced by the RDA reaction. Meanwhile, they all displayed the characteristic neutral losses of 56 Da, indicating the presence of the prenyl group Thus, it was speculated that they were bavachin or isobavachin. Compared with compound **1**, compound **6** generated an ion of 219 Da as the base peak, indicating the presence of -OCH_3_ on the A ring. Compound **6** was thus considered bavachinin A (BVA). Based on the results of mass spectrometry analysis, compounds **1**–**8** were directionally isolated from Fr. A (Appendix A). Furthermore, the standard compounds were used to confirm the reliability of the developed approach based on UPLC-MS/MS (Figure 2B).

### 3.3. Effects of IPFs on MRSA Membrane

The effects of Fr. A on the morphological and ultrastructural changes of MRSA were observed by using SEM. Surprisingly, Fr. A exhibited an obvious influence on the morphology of MRSA cells. The bacteria in the control group (A) showed a clear and smooth surface. In contrast, MRSA cells treated by Fr. A displayed an irregular morphology on the outer surface and some pores (Figure 3B), indicating that Fr. A might damage the MRSA membrane.

The results of the antibacterial assay (Table 4) indicated that the target compounds in Fr. A showed potent antibacterial activity, especially against MRSA. Compared with the non-prenylated flavonoid, daidzein, it was concluded that the presence of a prenyl chain or a pyran ring on the substituent moiety could significantly enhance the antibacterial activities of isoflavones. Interestingly, it was observed that NBF showed better antibacterial activities than CL, indicating that increasing the length of the isopentenyl side chain could significantly improve the antibacterial activities of IPFs. Previous studies have demonstrated that the presence of the isopentenyl group could significantly improve the antibacterial activities of chalcones against MRSA [24,25,26]. Isobavachin showed the best antibacterial activities among isobavachin, bavachin, and naringenin, indicating that the presence of isopentenyl and hydroxyl groups could significantly improve the antibacterial activities of flavanones.

To further investigate the effect of IPFs isolated from Fr. A on the MRSA membrane, the PI fluorescence probe was used to evaluate membrane permeability. PI is membrane-impermeable and can only enter the cytoplasm and bind to DNA when the cell membrane has been disrupted [27]. By measuring the intensity of fluorescence, we can show the integrity of the cytoplasmic membrane and the ability of the IPFs to permeabilize the membrane. Compared with the control group, all the IPFs could significantly increase the fluorescence intensity of PI, indicating that the IPFs could cause the loss of MRSA membrane integrity (Figure 4A). To further confirm that IPFs could specifically bind to the MRSA membrane, the effects of the components of the bacterial membrane on the activities of IPFs against MRSA were evaluated. The exogenous addition of PG and CL, which are the major component in the bacterial membrane, could decrease the activity of IPFs in a dose-dependent manner (Figure 4C–G), indicating that the exogenous addition of membrane phospholipid can effectively promote bacterial repair of the cell membrane. Taken together, the presence of isopentenyl is essential for the antibacterial activities of flavonoids, and such an effect may be achieved by targeting the cell membranes. Therefore, three different types of IPFs: BVC (flavanone), IBC (chalcone), and CLA (isoflavone) were selected for the further study of the related mechanism against MRSA.

### 3.4. Mechanism of IPFs against MRSA

The CM plays a key role in bacterial growth and survival because the damage to the CM could cause a series of negative effects on the bacteria, including ROS production, changes in the proton motive force, and membrane fluidity [28]. It was reported that the insertion of membrane-disrupting antibacterial agents into lipid bilayers could cause dramatic changes in membrane fluidity, resulting in the dysfunction of membranes such as leakage of intracellular components and loss of membrane protein functions [29,30]. Therefore, the laurdan fluorescence probe was used to determine the effects of IPFs on the MRSA membrane’s fluidity. As shown in Figure 5A, IBC at a concentration of 1 MIC could significantly increase the membrane fluidity, while BVC and CLA could decrease the membrane fluidity. In line with that, the change of membrane fluidity disrupts cellular homeostasis, resulting in the disorder of the membrane potential (Δ*ψ*) and osmotic pressure. Subsequently, we measured the Δ*ψ* by a fluorescent DiSC3(5). As shown in Figure 5B, the fluorescence intensity significantly decreased with the treatment of IBC, BVC, or CLA at 1–4 MICs, indicating that the Δ*ψ* of MRSA was significantly depolarized. Pathogenic bacteria develop resistance because bacterial cells produce biofilms that make antibiotics impenetrable. Therefore, the ability to penetrate the biofilm and target the cytoplasmic membrane (CM) is the key to solving the problem of multidrug resistance [31,32]. Here, we found no de novo resistance to IBC, BVC, or CLA appearing during the 20-day serial passage of MRSA (Figure 5C–E), indicating that they could directly interact with the CM.

The depolarization of the membrane potential might lead to membrane fragmentation and cell lysis. Consequently, disrupted membrane homeostasis may cause oxidative stress and the accumulation of ROSs [33,34]. We measured the ROS production level with a DCFH-DA fluorescence probe. The results showed that IBC or BVC treatment contributed to the excessive accumulation of intracellular ROSs in MRSA in a dose-dependent manner (Figure 6A). When the membrane is injured, changes in the permeability of the cell membrane may cause the leakage of soluble cellular substances such as nucleic acids, proteins, ATP, etc. [35,36,37]. Consistent with this, we found that the extracellular ATP level of MRSA gradually increased with the increase of the IBC, BVC, or CLA concentration (Figure 6B). Furthermore, the affinity between POPG and CLA, which showed the best antibacterial activities against MRSA, was determined and calculated by using an ITC assay (Figure 6C), where the equilibrium dissociation constant (K_D_), number of binding sites (N), and molar binding enthalpy (ΔH) were 0.109 mol L^−^^1^, 1.14, and −9.91 kJ mol^−^^1^, respectively. These results illustrated that the interaction between IPFs and the MRSA membrane was responsible for their potent activities against MRSA.

The biophysical integrity and function of the CM are critical for bacterial survival. Therefore, antibacterial agents targeting the bacterial CM are promising therapeutics [31,32,38]. In this study, we found that IPFs in PCSs showed strong disruption to the CM of MRSA. Since the presence of the isopentenyl side chains, IPFs showed strong hydrophobicity. Therefore, they can easily penetrate through the lipid bilayer membrane, affecting the proportion of unsaturated fatty acids in the cell membrane and altering its structure. When the integrity of the CM is broken, some cytoplasmic inclusion such as betaine, choline, etc., which are generic osmotic protective substances, will leak out of the cell and cause changes in the osmotic pressure of cells. In addition, the depolarization of the membrane potential and alteration of osmotic pressure might lead to membrane fragmentation and cell lysis [39,40,41]. Consequently, disrupted membrane homeostasis may cause oxidative stress and the accumulation of ROSs. In this study, we measured the ROS production level by using a DCFH-DA fluorescence probe. IPF treatment contributed to the excessive accumulation of intracellular ROSs in MRSA in a dose-dependent manner. What is more, we observed an increasing level of the extracellular ATP, further suggesting that the membrane rupture occurred under the treatment of IPFs. Taken together, our findings indicated that the IPFs such as BI, IBC, and CLA could target the MRSA membrane, resulting in membrane dysfunction, including changes in the proton motive force and membrane fluidity, the accumulation of numerous intracellular ROSs, and the outflow of intracellular ATP, which further leads to cell death (Figure 6D).

Although we demonstrated the possible mechanism of IPFs in PCSs against MRSA, the pharmacodynamical behaviors still need further studies and optimization in the future. What is more, discovering the leading compounds for antibacterial drugs with similar structures and better activity in more plants needs further exploration.

## 4. Conclusions

In the present study, we developed a fast and reliable approach based on UPLC-MS/MS to screen out IPFs from PCS extracts. We also confirmed the IPFs from PCSs possessed potent activities against Gram ^(+)^ bacteria, especially against MRSA. Importantly, we illustrated that these compounds could target the MRSA membrane, causing membrane dysfunction and cell death. Taken together, this strategy can be applied to rapidly screen out IPFs with significant antibacterial activity from natural plants.

## Figures and Tables

**Figure 1 molecules-27-06952-f001:**
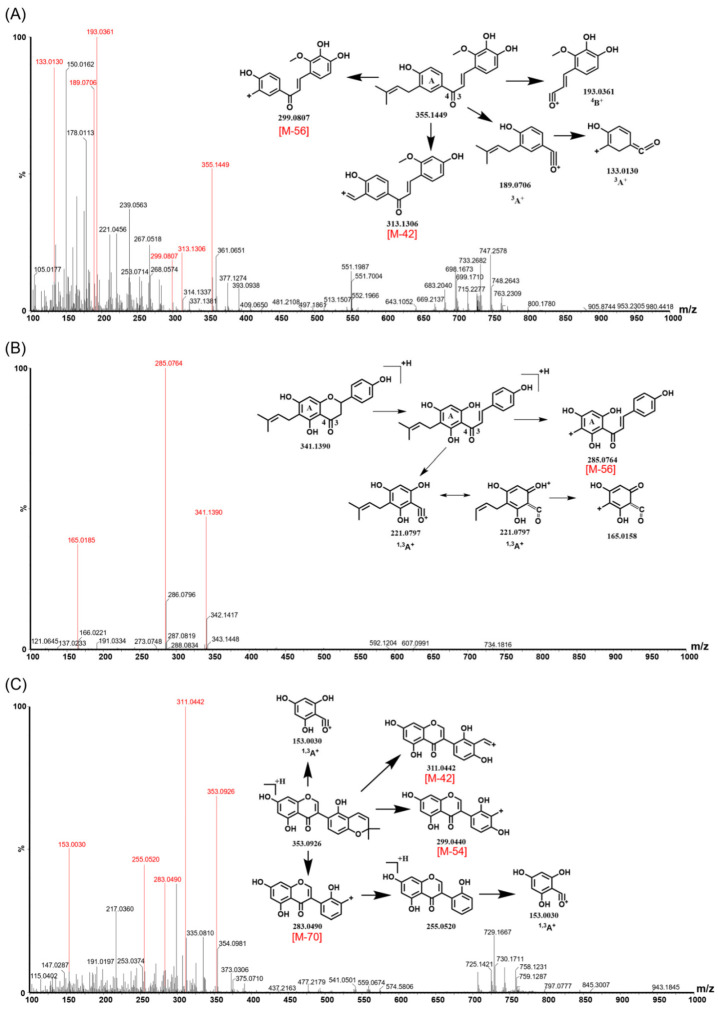
Typical fragment ions and the fragmentation pathway of reference flavonoids in PI mode. (**A**) Licochalcone D; (**B**) 6-prenylnaringenin; (**C**) licoisoflavone B.

**Figure 2 molecules-27-06952-f002:**
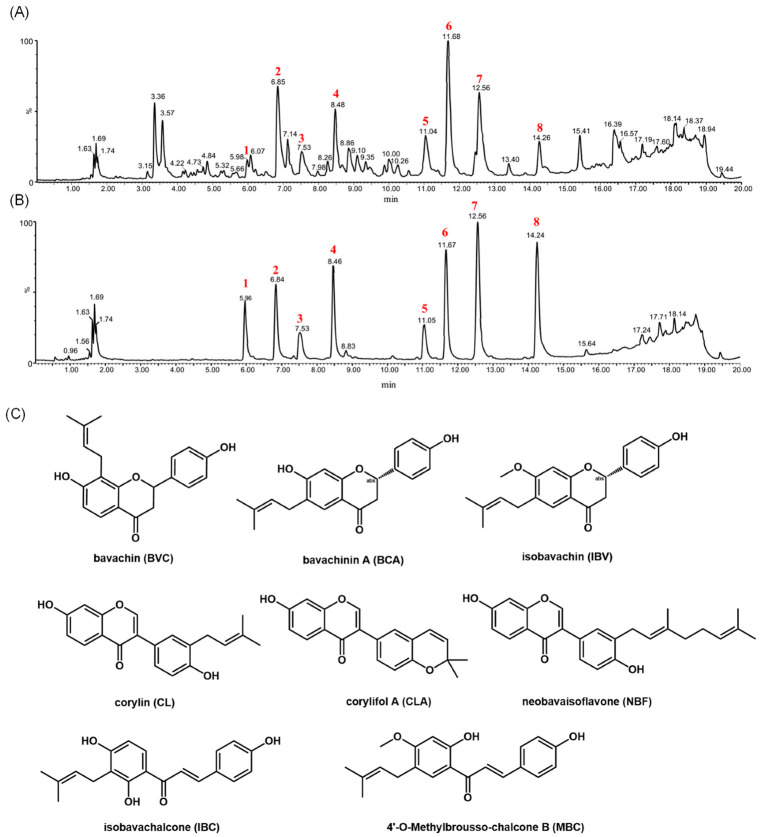
UPLC-MS/MS chromatogram of (**A**) Fr. A and (**B**) the references of compounds **1**–**8** in CPSs. (**C**) The chemical structure compounds **1**–**8** in Fr. A.

**Figure 3 molecules-27-06952-f003:**
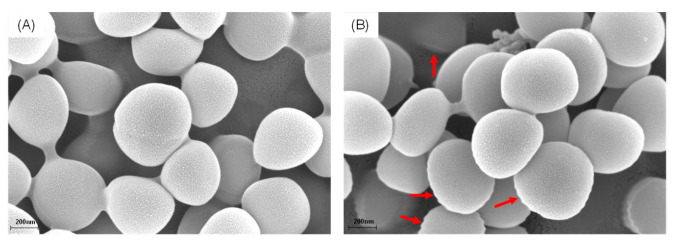
Bacterial morphology affected by Fr. A in the (**A**) control group and (**B**) treatment group.

**Figure 4 molecules-27-06952-f004:**
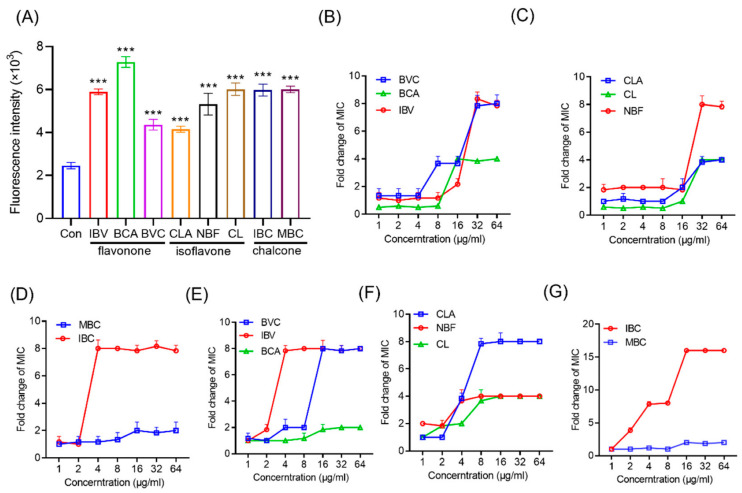
The IPFs in PCSs show robust antibacterial activity against MRSA. (**A**) Increased membrane permeability after treatment of IPFs in PCSs. (**B**–**G**) Exogenous addition of phosphatidylglycerol (PG) and cardiolipin (CL) abolished the antibacterial activity of flavonone (**B**,**E**), isoflavone (**C**,**F**), and chalcone (**D**,**G**) against MRSA. The concentrations of phospholipids were in the range of 1 to 64 µg mL^−1^. Values are presented as the mean ± SD. *p* < 0.05 was considered statistically significant, *** *p* < 0.01 vs. the control (Con) group.

**Figure 5 molecules-27-06952-f005:**
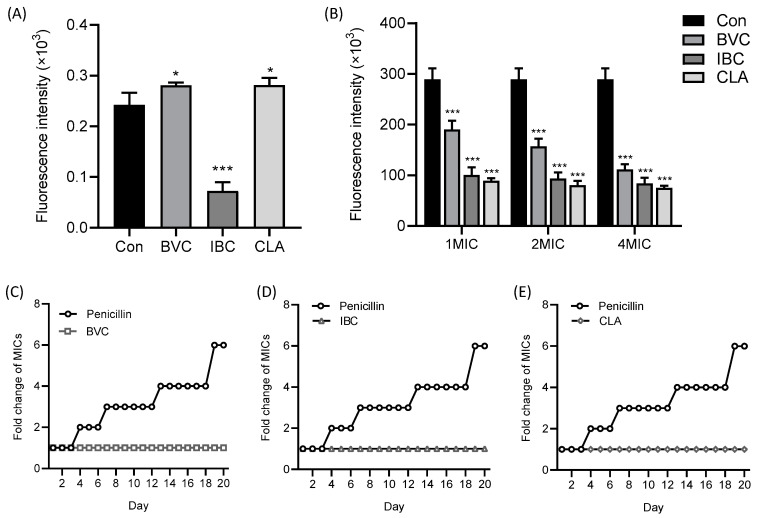
IPFs in PCSs exert antibacterial effects through membranes. (**A**) The fluidity of the membrane was significantly changed for MRSA after treatment with IPFs. (**B**) Membrane potential for MRSA after treatment with IPFs. No resistance to BVC (**C**), IBC (**D**), or (**E**) CLA occurred in 20-day serial passage. Values are presented as the mean ± SD. *p* < 0.05 was considered statistically significant, * *p* < 0.05, *** *p* < 0.01 vs. the control (Con) group.

**Figure 6 molecules-27-06952-f006:**
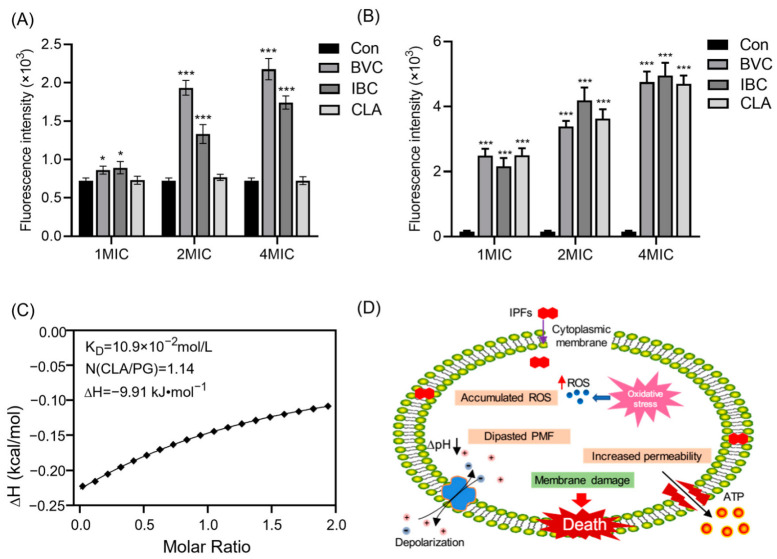
IPFs in PCSs exert antibacterial effects through membranes. (**A**) The effect of IPFs on the ROS accumulation in MRSA. (**B**) The extracellular ATP level of MRSA treated with IPFs. (**C**) Isothermal titration calorimetry (ITC) analysis of the interaction between PG and CLA. (**D**) Mechanism of action of IPFs in Gram-positive bacteria. Values are presented as the mean ± SD. *p* < 0.05 was considered statistically significant, * *p* < 0.05, *** *p* < 0.01 vs. the control (Con) group.

**Table 1 molecules-27-06952-t001:** The MS data of the reference compounds.

No.	Name	Formula	[M + H]	Typical Fragment Ions (Da)	Typical Losses
1	Licochalcone D	C_21_H_22_O_5_	355.1557	313.1306, 299.0807, 193.0301, 189.0760, 133.0130	42 Da, 56 Da
2	Licochalcone B	C_16_H_14_O_5_	287.0707	193.0251, 121.0025	
3	Glabrone	C_20_H_16_O_5_	337.0972	295.0490, 283.0486, 267.0648, 239.0527	28 Da, 42 Da, 54 Da, 70 Da
4	Licoisoflavone B	C_20_H_16_O_6_	353.0926	311.0442, 299.0442, 283.0490, 255.0520, 153.0030	42 Da, 54 Da, 70 Da
5	6-prenylnaringenin	C_20_H_20_O_5_	341.1390	285.0764, 221.0791, 165.0158	56 Da

**Table 2 molecules-27-06952-t002:** The MICs (μg/mL) of different fractions from PCS.

Strains	EE	EOE	Fr. A	Fr. B	Fr. C	Fr. D	Oxacillin
*Staphylococcus aureus*	125	62.5	31.2	125	250	250	31.2
*Staphylococcus epidermidis*	125	62.5	31.2	125	250	250	31.2
MRSA	125	62.5	31.2	125	250	250	31.2
*Shigella dysenteriae*	NA	NA	NA	NA	NA	NA	NA
*Pseudomonas aeruginosa*	NA	NA	NA	NA	NA	NA	NA
*Escherichia coli*	NA	NA	NA	NA	NA	NA	NA

**Table 3 molecules-27-06952-t003:** The MS data of the prenylated flavonoids in Fr. A.

No.	Name	Formula	[M + H]	Typical Fragment Ions (Da)	Typical Losses (Da)
1	Isobavachin	C_20_H_20_O_4_	325.1321	269.0677, 205.0691 149.0081	56
2	Neobavaisoflavone	C_20_H_18_O_4_	323.1176	267.0520, 255.0521 239.0563, 137.0080	56
3	Bavachin	C_20_H_20_O_4_	325.1312	269.0690, 205.0693 149.0079	56
4	Corylin	C_20_H_16_O_4_	321.1025	279.0534, 267.0527 251.0566, 137.0078	42, 54, 70
5	Isobavachalcone	C_20_H_20_O_4_	325.1332	269.0688, 205.0731 149.0081	56
6	Bavachinin A	C_21_H_22_O_4_	339.1493	283.0838, 219.0871 151.0236, 119.0341	56
7	Corylifol A	C_25_H_26_O_4_	391.1815	323.1172, 267.0531 239.0565, 137.0084	56
8	4′-O-Methylbrousso-chalcone B	C_21_H_22_O_4_	339.1479	283.0828, 271.0841 219.0872, 151.0241	56

**Table 4 molecules-27-06952-t004:** The MIC of IPFs and non-prenylated flavonoids against common Gram-positive bacteria.

Compound	MIC (μg/mL)		
	*S. aureus*	*S. epidermidis*	MRSA
Isobavachin (IBV)	15.6	7.80	7.80
Neobavaisoflavone (NBF)	15.6	7.80	7.80
Bavachin (BVC)	7.80	15.6	7.80
Corylin (CL)	15.6	15.6	15.6
Isobavachalcone (IBC)	15.6	7.80	7.80
Bavachinin A (BCA)	31.2	31.2	31.2
Corylifol A (CLA)	3.90	3.90	3.90
4′-O-Methylbrousso- chalcone B(MBC)	31.2	31.2	31.2
Daidzein	NA	NA	NA
Naringenin	NA	NA	NA

## Data Availability

Not applicable.

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
