# Peer review of "Exploration of Antimicrobial Ingredients in Psoralea corylifolia L. Seed and Related Mechanism against Methicillin-Resistant Staphylococcus aureus"

_molecules, 2022, doi:10.3390/molecules27206952_

Round 1
Reviewer 1 Report
The manuscript (molecules-1956077) entitled "Exploration of antimicrobial ingredients in Psoralea corylifolia L. seed and related mechanism against methicillin-resistant Staphylococcus aureus" is relevant in the present context, well designed and written with confidence. The main aim of this study was to develop a fast and reliable method for screening out antimicrobial compounds targeting MRSA membrane from Psoralea corylifolia L. seed. The results confirmed that the IPFs possessed potent activities against Gram (+) bacteria.
However, some suggestions against a few minor concerns are there which are to be addressed before considering the MS for its final publication:
1. Rewrite the sentence (Line #69-70) for clarity
2. Cite suitable reference (Line # 53-56)
3. Is there any reason to write in increased font size at line # 326?
4. Instead of two separate sentences, it should be merged to a single sentence at line # 343-344.
Author Response
- Rewrite the sentence (Line #69-70) for clarity
Reply: Thank you for your advice, we have rewritten the sentence.
- Cite suitable reference (Line # 53-56)
Reply: Thank you for your advice, we have replaced the suitable citation.
- Is there any reason to write in increased font size at line # 326?
Reply: Thank you for your advice, we have revised the font size at line 326.
- Instead of two separate sentences, it should be merged to a single sentence at line # 343-344.
Reply: Thank you for your advice, we have revised the sentence according to your advice
Reviewer 2 Report
1. Numerous reviews published in 2020-2021 as mentioned doi below, are not cited in the Introduction section.
https://doi.org/10.3390/ijms21239028, https://doi.org/10.3390/molecules26113447, https://doi.org/10.3390/molecules26103005
2. MIC citation was missing in the materials and methods section.
Author Response
- Numerous reviews published in 2020-2021 as mentioned doi below, are not cited in the Introduction section.
https://doi.org/10.3390/ijms21239028, https://doi.org/10.3390/molecules26113447, https://doi.org/10.3390/molecules26103005
Reply : Thank you for your advice, we have cited the newly published reviews in 2020-2021 according to your suggestion.
- MIC citation was missing in the materials and methods section.
Reply : Thank you for your advice, we have cited the MIC citation.
Reviewer 3 Report
The paper is well written, well organized and discussed. The overall merit of this investigation makes it suitable for publication in Molecules. Slight spelling erros are found in the text, however, the paper just requieres minor revision. Please address the following points:
1. English must be brushed up one more time.
2. Separate Figure 4A from the others. Actually, this section is a Table that should be presented in the correct format.
3. Add error bars in Figure 4C-F and 5C-E. Despite are MIC values, this parameter was obtained after several assays
4. Increase the size of the red numbers in each peak of the chromatogram in Figure 2.
5. The list of references should be carefully revised since there are several style errors. Please verify the instructions for authors
Author Response
- English must be brushed up one more time.
Reply: Thank you for your advice, we have improved our English writing according to your advice.
- Separate Figure 4A from the others. Actually, this section is a Table that should be presented in the correct format.
Reply: Thank you for your advice, we have separated Figure 4A as Table 4 following your suggestion
- Add error bars in Figure 4C-F and 5C-E. Despite are MIC values, this parameter was obtained after several assays
Reply: Thank you for your advice, we have added error bars in Figure 4 C-F and 5C-E
- Increase the size of the red numbers in each peak of the chromatogram in Figure 2.
Reply: Thank you for your advice, we have revised Figure 2 according to your advice.
- The list of references should be carefully revised since there are several style errors. Please verify the instructions for authors
Reply: Thank you for your advice, we have revised the references according to your advice.